# Scale-Specific Prediction of Topsoil Organic Carbon Contents Using Terrain Attributes and SCMaP Soil Reflectance Composites

**Markus Möller** [1,*] , **Simone Zepp** [2] , **Martin Wiesmeier** [3] , **Heike Gerighausen** [1] and **Uta Heiden** [4]

1 Julius Kühn Institute (JKI), Federal Research Centre for Cultivated Plants, Institute for Crop and Soil Science, Bundesallee 58, 38116 Braunschweig, Germany; heike.gerighausen@julius-kuehn.de
2 German Aerospace Center (DLR), German Remote Sensing Data Center (DFD), Münchener Str. 20, 82234 Wessling, Germany; simone.zepp@dlr.de
3 Bavarian State Research Center for Agriculture, Institute for Organic Farming, Soil and Resource Management, Lange Point 6, 85354 Freising, Germany; martin.wiesmeier@lfl.bayern.de
4 German Aerospace Center (DLR), Remote Sensing Technology Institute (IMF), Münchener Str. 20, 82234 Wessling, Germany; uta.heiden@dlr.de
* Correspondence: markus.moeller@julius-kuehn.de

**Abstract:** There is a growing need for an area-wide knowledge of SOC contents in agricultural soils at the field scale for food security and monitoring long-term changes related to soil health and climate change. In Germany, SOC maps are mostly available with a spatial resolution of 250 m to 1 km$^2$. The nationwide availability of both digital elevation models at various spatial resolutions and multi-temporal satellite imagery enables the derivation of multi-scale terrain attributes and (here: Landsat-based) multi-temporal soil reflectance composites (SRC) as explanatory variables. In the example of a Bavarian test of about 8000 km$^2$, relations between 220 SOC content samples as well as different aggregation levels of the explanatory variables were analyzed for their scale-specific predictive power. The aggregation levels were generated by applying a region-growing segmentation procedure, and the SOC content prediction was realized by the Random Forest algorithm. In doing so, established approaches of (geographic) object-based image analysis (GEOBIA) and machine learning were combined. The modeling results revealed scale-specific differences. Compared to terrain attributes, the use of SRC parameters leads to a significant model improvement at field-related scale levels. The joint use of both terrain attributes and SRC parameters resulted in further model improvements. The best modeling variant is characterized by an accuracy of $R^2 = 0.84$ and $RMSE = 1.99$.

**Keywords:** soil reflectance composites; digital soil modeling; soil organic carbon; GEOBIA; Landsat; terrain analysis

## 1. Introduction

Soil is the largest carbon sink on earth after the oceans and can store more than twice as much $CO_2$ as the atmosphere [1]. Therefore, the soil of agricultural ecosystems can contribute to the mitigation of greenhouse gas (GHG) emissions and thus to climate change mitigation through increased carbon sequestration [2]. In order to assess this potential and promote it through adaptation of land use systems, as well as to localize adaptation needs on an area-by-area basis in the context of the European Common Agricultural Policy (CAP) and the Sustainable Development Goals (SDGs), up-to-date, area-wide, and high-resolution information on carbon contents of agricultural soils is needed [3,4]. Germany-wide maps of the carbon content of agricultural soils are currently only available as static maps with a spatial resolution of 200 m$^2$ to 1 km$^2$ [5]. The maps are not suitable as a basis for small-scale field-specific analyses. In addition, the maps do not contain quality measures that are important for communicating model uncertainties [6,7].

Detailed information on carbon content is available in the form of point soil samples collected at the state, national or European level (e.g., [8–10]). The data sets differ in sampling methodology, frequency, and density as well as in their representativeness. Explanatory variables are needed for the operational transformation of point data into spatial data sets. Since the nationwide availability of digital elevation models at various spatial resolutions, digital soil mapping has transitioned from the research phase to operational use [11]. The increasing availability of multi-temporal satellite imagery allows an expansion of the data space to distinguish both spatial and temporal patterns of SOC content [4,12]. Multi-temporal soil reflectance composites (SRC), based on Landsat or Sentinel-2 time series, have proven as explanatory variables for the prediction of (top)soil organic carbon (SOC) content [5,13–20].

In this article, we deepen a study by Zepp et al. (2021), which applied different modeling methods on Landsat-based SRC data for SOC content prediction in Bavaria, Germany [5]. As a result, the Random Forest (RF) showed the best predictive capabilities in terms of model accuracy and performance. Using a sub-area within Bavaria as an example, we extend the modeling approach and compare the predictive single and mutual capabilities of Landsat-based SRCs and multi-scale terrain attributes. The latter should take into account the fact that soil properties and soil-forming processes are an expression of complex relationships between soil forming factors and landforms, which occur on different scales [21–28]. Multi-scale terrain attributes enable the consideration of contextual information, which can improve the prediction accuracy of soil properties [24]. In addition, different aggregation levels of the two parameter sets were generated. The used segmentation algorithm results in spatial objects with soil-related meaning [29,30], which are also referred as soil-terrain objects [6] or ecotopes [31]. They can be defined as groups of terrain attribute raster cells which are aggregated to landform elements according to a scale-specific homogeneity [29,32,33]. Their usage in digital soil mapping applications has been proven as superior compared to pixel-based approaches [30,34,35]. The main objective of this study is the analysis of relations between SOC content samples as well as different aggregation levels of terrain attributes and Landsat-based SRC data regarding their scale-specific predictive power.

## 2. Materials and Methods

Figure 1 illustrates the principle digital soil modeling workflow, in which scale-specific reference units (RU) with explanatory variables are related to soil measurements and analyzed with machine learning methods. The workflow can be distinguished into the two categories:

1. "Input data" comprises the provision of soil samples (Section 2.1.1) as well as the derivation of terrain attributes (Section 2.1.2) and multi-temporal SRCs (Section 2.1.3). By applying a segmentation algorithm (Section 2.1.4), both data types are used for generating multi-hierarchical reference units (RU), which are parameterized by applying zonal statistics operations (Section 2.1.5).
2. "Machine learning" refers to the actual spatial SOC content prediction by applying the Random Forest algorithm. In addition, an internal and independent validation schema, as well as a recursive feature elimination analysis, is included (Section 2.2).

The workflow was implemented using R functions [36] documented in a Github repository (https://github.com/FLFgit/ScaleP.git; accessed on 13 March 2022).

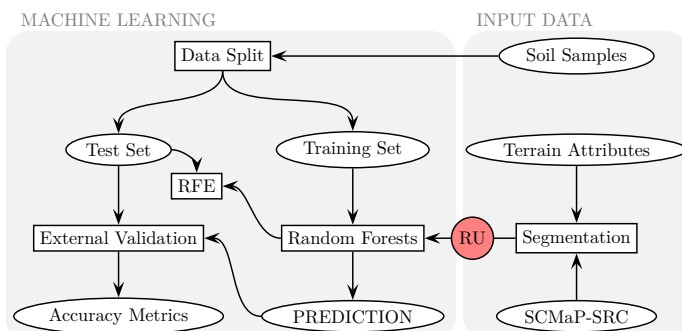

**Figure 1.** Workflow for the scale-specific SOC content prediction based on SCMaP Soil Reflectance Composites (SCMaP-SRC) and terrain attributes (RU: reference units; RFE: recursive feature elimination).

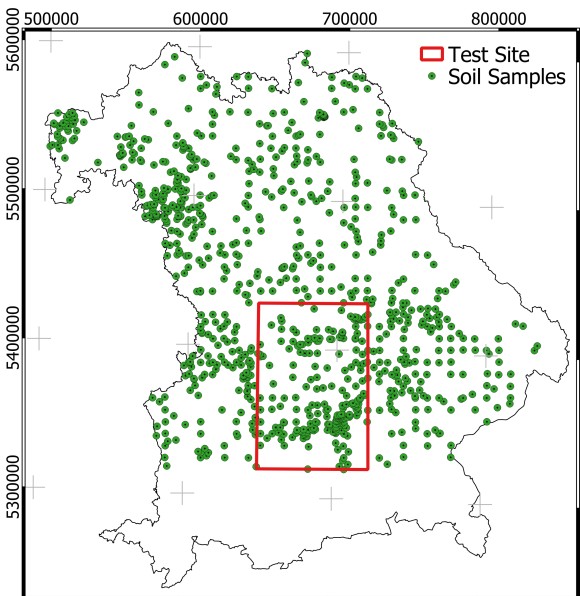

**Figure 2.** Test site location in Bavaria and the distribution of soil samples. Projection: EPSG 31468 (https://spatialreference.org/ref/epsg/31468; accessed on 13 March 2022).

*2.1. Input Data*

2.1.1. Soil samples

The test area in Bavaria (Figure 2) was selected because the ranges of SOC content values are comparable with the entire area (Figure 3). The data set comprises soils with well-developed B horizons (mainly Cambisols), soils with initial soil formation (mainly Leptosols), soils with water stagnation (mainly Stagnosols and Planosols), soils with clay migration (mainly Luvisols), clay-rich soils (mainly Vertisols), groundwater soils (mainly Gleysols), and natural bogs and fens (mainly Histosols) according to the German soil systematic and the equivalent reference soil groups of the WRB system [37]. Both mineral soils with lower SOC contents and organic soils in the form of fens (e.g., Königsmoos) with higher SOC contents occur in the test area. In addition, the test area has a comparable heterogeneous terrain composition as in Bavaria.

Figure 2 shows the spatial distribution of available sampling sites throughout Bavaria ($N = 939$) and in the test area of about $8000 \, \text{km}^2$ ($N = 220$) selected for modeling, with the size of the agricultural area accounting for about half. The available soil samples were provided by the Bavarian Environment Agency (LfU) and the Bavarian State Research Center for Agriculture (LfL). All databases were determined by dry combustion using elemental analyzers [8,9]. The final soil data set of the test site comprises 220 soil samples (LfL: 14 samples; LfU: 206 samples). The SOC contents range from 0.74 % to 18.3 % with

a median content of 2.00 %. The comparison of both value distributions reveals that differences mainly occur in the value range between 3 and 10 % (Figure 3). Applying the nonparametric Kolmogorov–Smirnov goodness-of-fit test [38] revealed that the empirical cumulative distributions are significantly different, with the $D = 0.24$ of the curves being rather small (cf., [6]).

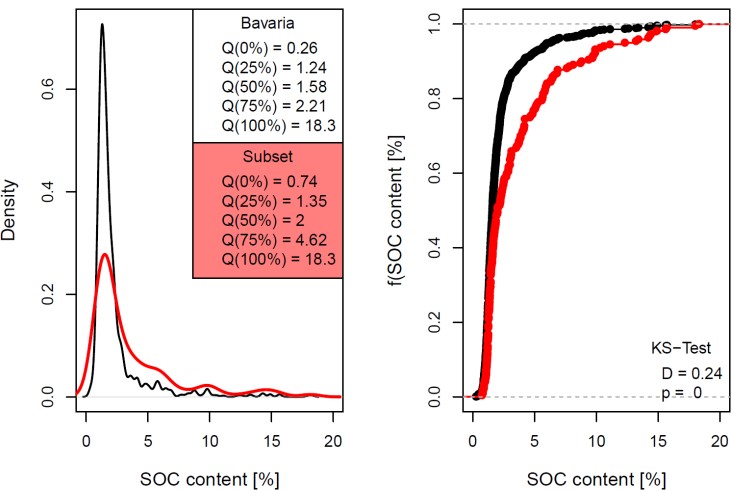

**Figure 3.** Comparison of soil samples' SOC content [%] distributions: density plot with quantile values *Q* (**left**) and plot of empirical cumulative distribution functions (ECDF) with ECDF distance *D* (**right**) for entire Bavaria (black) and the subset (red; cf., Figure 2).

### 2.1.2. Terrain Attributes

Terrain attributes have been established for decades as explanatory variables for predicting soil properties in general [11,39] and SOC content in particular [40]. Since the scale dependency of terrain attributes has an effect on SOC predictions (e.g., [23,26]), multi-scale terrain attributes have been derived (Table 1). This concerns in particular variants of attributes "Normalized Height" (*NH*; Figure 4a,b) and "Topographic Position Index" (*TPI*) [41,42], for whose calculation different moving window sizes were applied. The variants of the attributes "Vertical Distance above Channel Network" (*VDC*) and "Terrain Classification Index" (*TCI*) [41] are based on different aggregation levels of the channel network derivations. The "Mass Balance Index" (*MBI*) versions are expressions of the differentiability regarding dominant and subdominant relief shapes [21]. All multi-scale variants are based on tuning parameters listed in Table 1. Their definition is the result of exponential functions for which the start and end values were determined empirically (see https://github.com/FLFgit/ScaleP/blob/master/callScaleP.R; accessed on 13 March 2022). Finally, the one-dimensional attributes including local attributes (sink-filled digital elevation model (*FILL*) and "Slope" (*SLP*)) as well as regional attributes ("Topographic Openness" (*TOP* and *TON*) [43] and the "Topographic Wetness Index" (*TWI*; Figure 4c) [44]) were calculated [45].

The corresponding process chain is documented in an R function (Table 1). There, all terrain attribute variants are defined. In this study, a DEM with a resolution of 10 m was used, which is provided by the German Federal Agency of Cartography and Geodesy (https://gdz.bkg.bund.de; accessed on 13 March 2022). The DEM was resampled to 30 m according to the resolution of the SCMaP SRC data set (Section 2.1.3).

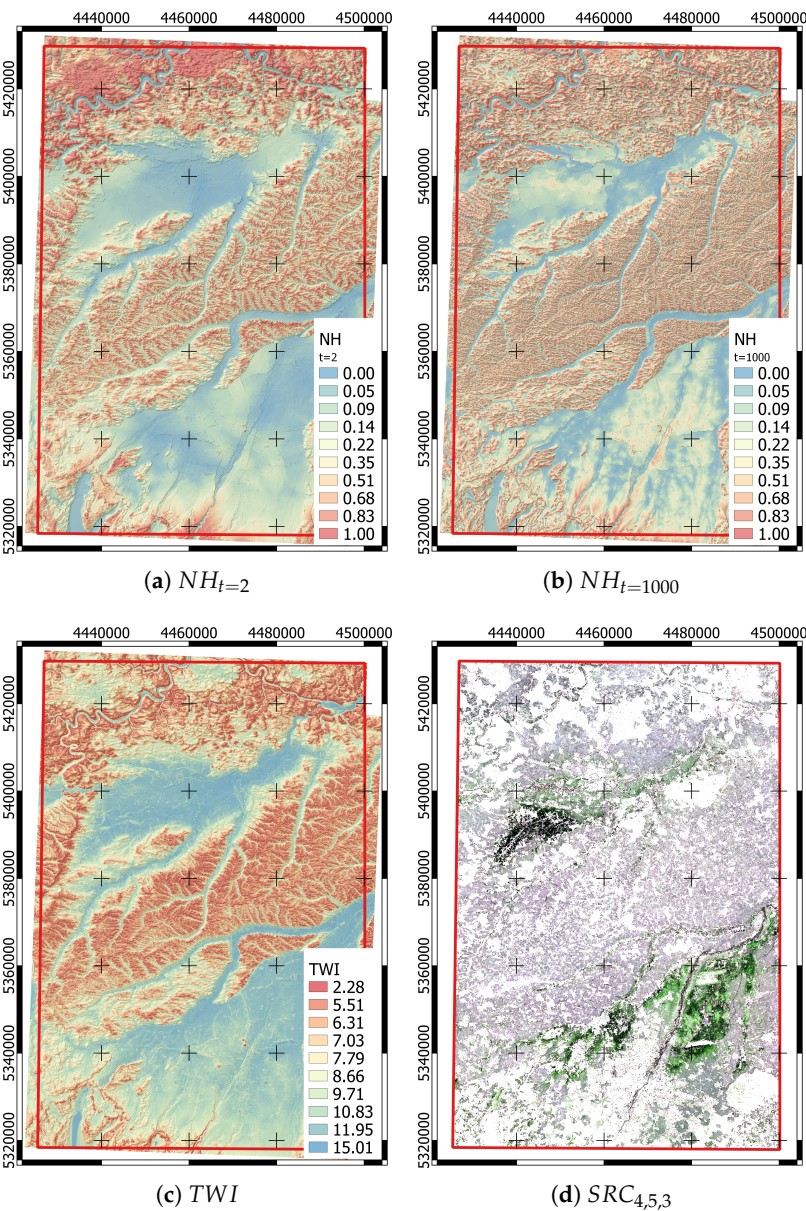

**Figure 4.** Visualization of multi-scale (**a**,**b**) and one-dimensional terrain attributes (**c**); cf., Table 1) as well as of selected SRC bands (**d**). Projection: EPSG 31468 (https://spatialreference.org/ref/epsg/31 468; accessed on 13 March 2022).

**Table 1.** Explanatory variables for the SOC content prediction: terrain attribute variants and SCMaP-SCR bands ($SM_{1-...}$: definitions of terrain attribute variants and tuning parameters see function fNumP() and R function collection (https://github.com/FLFgit/ScaleP/blob/master/callScaleP.R; accessed on 13 March 2022).

| Explanatory Variable | Meaning | Multi-Scale Tuning Parameter (Start and End Value) | Variant Number | Source |
|---|---|---|---|---|
| $FILL$ | Digital Elevation Model with filled sinks | – | 1 | [46] |
| $SLP$ | Slope | – | 1 | [47] |
| $VDC$ | Vertical Distance above Channel Network | Catchment Area (CA $\in$ [10,000:1000,000]) | 10 | [41] |
| $TCI$ | Terrain Classification Index | Catchment Area (CA $\in$ [10,000:1000,000]) | 10 | [41] |
| $TWI$ | Topographic Wetness Index | – | 1 | [44] |
| $MBI$ | Mass Balance Index | Curvature Transfer Constant ($T \in$ 0.0001:0.1) | 10 | [21] |
| $TOP$ | Topographic (positive) Openness | – | 1 | [43] |
| $TON$ | Topographic (negative) Openness | – | 1 | [43] |
| $NH$ | Normalized Height | Generalization Parameter ($t \in$ [2:1000]) | 10 | [41] |
| $TPI$ | Topographic Position Index | Scale Parameter ($S \in$ [20:1000]) | 10 | [42] |
| $SRC_{1-7}$ | SCMaP-SRC (1984–2014), Landsat Reflectances | – | 7 | [48] |
| $SRC_{8-14}$ | SCMaP-SRC (1984–2014), normalized Landsat Reflectances | – | 7 | [48] |

### 2.1.3. SCMaP-SRC

In addition to the terrain attributes, spectral information of a remote sensing soil reflectance composite (SRC) was used for SOC content modeling (Figure 4d). The Soil Composite Mapping Processing (SCMaP) chain enables the generation of SRC for individually selected regions and time periods [48]. Based on a modified vegetation index (PV) two thresholds are determined to separate predominantly uncovered soils from all other land cover types. The development of the database for the threshold derivation is automated. The threshold itself has been derived based on manually defined percentile measures [49].

A 30-year (1984–2014) compositing period was chosen to enable a smooth spectral database [5]. The compositing period was chosen according to the dates of soil sampling. The 30-year SRC was built on all Landsat (69 Landsat-4 TM, 1784 Landsat-5 ETM, and 998 Landsat-7 ETM+) collection scenes [50] available between 1984 and 2014 with a resolution of 30 m for the investigation area. For all scenes, the same pre-processing steps were applied. The FMask algorithm was used to detect and remove clouds, cloud shadows, and pixels covered by snow [51,52]. Additionally, an atmospheric correction was performed using the Atmospheric Correction (ATCOR) software for satellite imagery [53]. The reflectance soil composites show the averaged reflectance per-pixel composites for the observed time period of exposed soils. The patterns in the reflectance soil composite correspond to patterns of existing soil maps and the underlying geological structural region. Products therefore provide useful information on soils and exposed soil coverage. The resulting bands $SRC_{1-7}$ represent the "normal" averaged reflectances, for bands $SRC_{8-14}$ the averaged reflectances are normalized per scene by the albedo, which is calculated as mean reflectance over all six reflective bands.

### 2.1.4. Segmentation

Different aggregation levels of explanatory variables were derived by applying a region-growing segmentation algorithm that spatially groups grid cells of terrain attributes and SRC bands according to their neighborhood in a feature space and raster. As a result, polygons with scale-specific comparable heterogeneity are generated.

The region-growing algorithm "Fractal Net Evolution Approach" (FNEA) was applied [54], which is implemented in the software eCognition and has been shown to be suitable for detecting spatial objects of soil-related relevance [6,30,34,35] in the context of geographic object-based image analysis (GEOBIA) [55–57]. The algorithm relies on seed pixel groups with the smallest (here: Euclidean) distance in both the raster and the feature space of the parameters used. Then, the seeds grow until the user-specific heterogeneity of the raster values within the resulting objects is reached.

In this study, the segmentation input data were the variables $TWI$, $SLP$, $TCI_{CA=10000}$, $MBI_{T=0.0005}$, $NH_{t=1000}$ as well as $SRC_{8-14}$ (cf., Table 1). The shape of the resulting objects is influenced by the user-defined parameters "shape variance" and "compactness", which have been set to 0.05 and 0.01 here. The degree of object aggregation is controlled by the parameter "Scale Level" $L$ (cf., [30]). The corresponding 17 scale level values are listed in Table 1. As an example of a test site subset, Figure 5a displays polygons of three scale levels. There, red-colored polygon boundaries represent parent polygons, which are decomposed by the smaller yellow-colored child and blue-colored grandchild polygons. The latter can be viewed as vectorized raster cells [21,30]. The white areas were identified by the SCMaP algorithm as areas of no or little reflectance changes. This applies, for example, to forests or built-up areas.

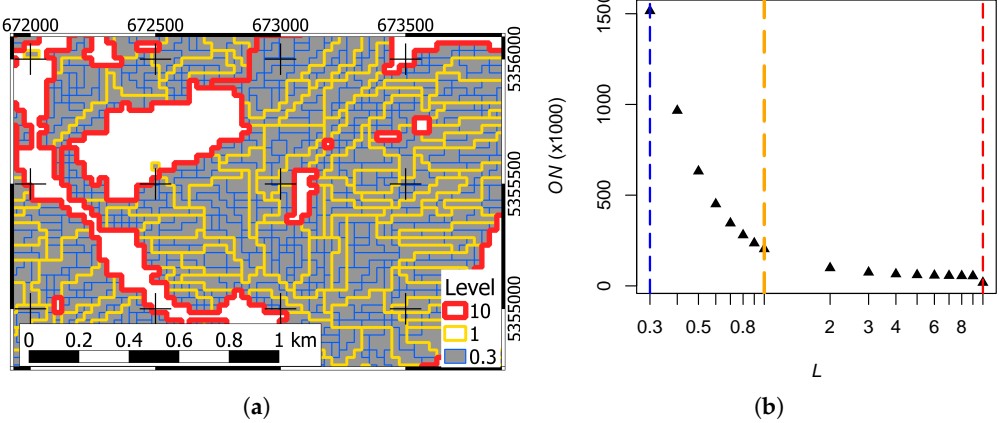

|        |        |
|:------:|:------:|
| (**a**) | (**b**) |

**Figure 5.** Visualization of reference unit-specific scale levels on the example of a test site subset (**a**) as well as relation between logarithmized scale level $L$ and object number $ON$ ((**b**); cf., Table 3). The colored dashed lines refer to the scale levels $L = 0.3$, $L = 1$ and $L = 10$. (**a**) White areas represent land that is not used by agriculture.

### 2.1.5. Parametrization

While the parametrization of scale-specific reference units (RU) is realized by applying zonal statistics functions [41,58], Zepp et al. (2021) applied a filter-based parameterization of the samples [5]. This involved averaging the spectra of the sample pixel and its eight neighboring pixels to reduce local and spatial variability. The background to this approach is that the SCMaP-SRC provides ground reflectance information at a pixel resolution of 30 m based on Landsat imagery. Linking point samples to a 30 m remote sensing pixel is a potential source of inaccuracy because not all samples are collected at least 30 m inside the fields. As a result, the SRC pixel may combine multiple surfaces with different spectral information. The ground sample is then combined with a mixture of spectral information. Therefore, spectral and spatial filtering was applied to the sample pixel and

its eight neighboring pixels to evaluate spectral differences within a field cluster. All pixel clusters showing deviating spectra are excluded from further processing.

### 2.2. Machine Learning

Random Forests (RF) is a regression and ensemble-based decision tree algorithm [59], which has been regularly used for predicting soil properties [60,61]. RF divides the feature space of the explanatory variables until the resulting tree has the best statistical correlation by minimizing the variance. Based on bootstrap samples, RF generates a large number of independent trees (ensembles). Two-thirds of the samples are used to grow the trees (in-bag data), and one-third are drawn randomly to calculate error estimates through cross-validation (out-of-bag data).

To validate the classification results, the total data set is randomly divided into a training and test data set of 75% and 25%, respectively, taking into account the target parameter distribution. Model building is based on the training data set. On the basis of the training data set, a calibration and a repeated 5-fold cross-validation are performed. The test data set is used for independent validation [62], to which the trained model is applied. Modeling performance is evaluated using the metrics of "Root Mean Square Error" ($RMSE$) and "Coefficient of determination" ($R^2$) for cross-validation, calibration, and independent validation. The $SLOPE$ of the regression line indicates the degree of underestimation or overestimation.

All used explanatory variables are more or less affected by multicollinearity. This concerns in particular the terrain attribute variations. RF can be considered tolerant of this phenomenon regarding the model prediction or the accuracy of the model [61]. However, collinearity might impair the interpretability of the model and may lead to misidentification of relevant predictors [63]. This especially concerns the interpretation of the variable importance of each explanatory variable, which is derived from the percent increase in mean squared error ($MSE$) resulting from the permutation of the out-of-bag data for each variable [64]. To ensure the interpretability of relevant predictors, the recursive feature elimination approach (RFE) is used [65,66], where the least important predictors are iteratively eliminated before the model is rebuilt [64].

## 3. Results

### 3.1. Filter-Based Parametrization

Table 2 lists the accuracy metrics resulting from applying the modeling approach to the samples for Bavaria [5] and the test site based on the filter-based parameterization. Figure 6 visualizes the corresponding validation scatter plots. The accuracy metrics include all $RMSE$ and $R^2$ values of calibration (CAL), cross-validation (CV), and (independent) validation (VAL) (Section 2.2).

**Table 2.** Accuracy metrics for the SOC content (%) prediction based on SCMaP-SRC parametrization applied by Zepp et al. (2021) [5] for Bavaria and the test site subset.

| Variant | Sample Number | $RMSE_{CV}$ | $R^2_{CV}$ | $RMSE_{CAL}$ | $R^2_{CAL}$ | $RMSE_{VAL}$ | $R^2_{VAL}$ | $SLOPE_{VAL}$ |
|---|---|---|---|---|---|---|---|---|
| Bavaria | 939 | 1.29 | 0.62 | 0.54 | 0.94 | 1.32 | 0.65 | 0.58 |
| Subset | 220 | 2.30 | 0.60 | 1.00 | 0.92 | 2.11 | 0.74 | 0.74 |

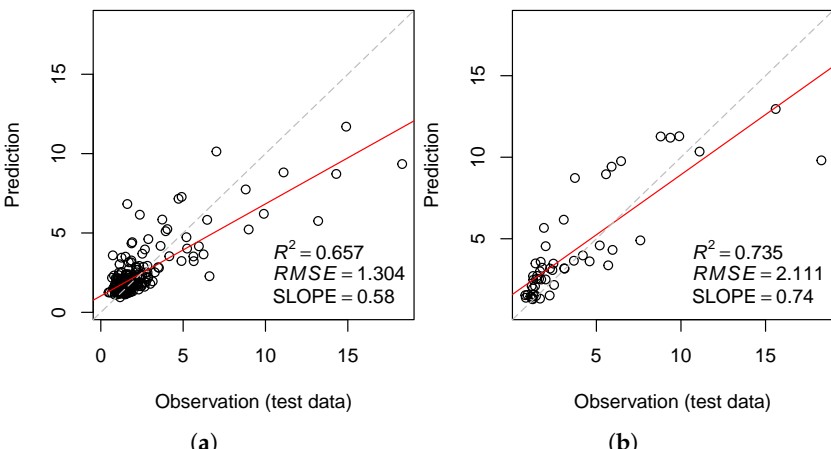

**Figure 6.** Comparison of independent validation results based on SOC content (%) data sets for entire Bavaria (**a**) and the subset ((**b**), cf., Table 2). The red line is the regression line of the scatter plot, the gray and dashed line indicates the optimal regression line.

The different accuracy metrics reflect the differences between the distributions of SOC content values, which is also indicated by the Kolmogorov–Smirnov distance between the entire Bavarian and the subset data set used in this study (Section 2.1.1). The differences mainly concern the $RMSE_{CV,CAL,VAL}$ and $R^2_{VAL}$ values, with the metrics of the subarea being higher than those for all of Bavaria.

### 3.2. Scale-Specific Parametrization

Table 3 summarizes all scale-specific accuracy metrics. The naming of the aggregation levels follows the scale level specification within the eCognition software setting, which was varied during the application of the algorithm and controls the parameter heterogeneity and thus indirectly the number and sizes of the resulting polygons [21,30,67]. Figure 5b and Table 3 show the relation between object number $ON$ and scale level $L$. From the size of the agricultural area (about 4000 km$^2$) and the number of objects, we find approximately the average object size, which varies depending on the heterogeneity of the soil landscape [6,21].

The CAL, CV, and VAL-related accuracy metrics are derived for all scale levels and the parameter variants "terrain attributes" (A), "SCMaP-SRC" (B), and "terrain attributes + SCMaP-SRC" (C). The CAL and VAL value differences reflect the degree of model overfitting [68], which correlates with the VAL values. It is also noticeable that the $R^2_{CV}$ values are often considerably smaller than $R^2_{VAL}$ values, which were also observed by Zepp et al. (2021) [5].

Figure 7 reveals scale-specific dependencies. Accordingly, the variation range of $R^2_{A,VAL}$ values (0.43 to 0.62) is smaller than of $R^2_{B,VAL}$ (0.47 to 0.84) and $R^2_{C,VAL}$ values (0.43 to 0.84). The $RMSE_{VAL}$ values vary between 2.78 and 3.47 (A), 2.00 and 3.28 (B), and 1.99 and 3.34 (C). With $R^2_{B,VAL} = 0.65$ and $RSME_{B,VAL} = 4.75$, the highest $R^2_{A,VAL}$ value is associated with scale level $L = 6$.

The use of SCMaP-SRC parameters leads to a significant model improvement. Two main scale ranges can be distinguished, with scale level 3 as the boundary. The highest $R^2_{B,VAL}$ value is found at scale level $L = 0.6$. The joint use of terrain attributes and SCMaP-SRC parameters leads to further slight model improvements. At the scale level $L = 0.3$, which corresponds to the original grid resolution of 30 m$^2$, the accuracy metrics again drop significantly. The $RMSE_{VAL}$ and $1 - SLOPE_{VAL}$ values display similar characteristics with opposite minima and maxima.

**Table 3.** Scale-specific accuracy metrics for SOC content (%) prediction variants based on terrain attributes (subscript A; cf. Table 1), SCMaP-SRC (subscript B) and both data sets (subscript C). *RMSE*—root mean square error; $R^2$—coefficient of determination; subscript CV—cross validation; subscript CAL—calibration; subscript VAL—independent validation. The gray and bold emphasized values refer to the prediction variant with highest accuracy metrics (Figure 9 and the corresponding scatter plots (Figure 8). The gray, bold and red values emphasized is related to the best prediction variant (cf., Figure 8f).

| Scale Level | Object Number | $RMSE_{A,CV}$ | $R^2_{A,CV}$ | $RMSE_{A,CAL}$ | $R^2_{A,CAL}$ | $RMSE_{A,VAL}$ | $R^2_{A,VAL}$ | $SLOPE_{A,VAL}$ | $RMSE_{B,CV}$ | $R^2_{B,CV}$ | $RMSE_{B,CAL}$ | $R^2_{B,CAL}$ | $RMSE_{B,VAL}$ | $R^2_{B,VAL}$ | $SLOPE_{B,VAL}$ | $RMSE_{C,CV}$ | $R^2_{C,CV}$ | $RMSE_{C,CAL}$ | $R^2_{C,CAL}$ | $RMSE_{C,VAL}$ | $R^2_{C,VAL}$ | $SLOPE_{C,VAL}$ |
|---|---|---|---|---|---|---|---|---|---|---|---|---|---|---|---|---|---|---|---|---|---|---|
| 10 | 18183 | 2.89 | 31 | 0.43 | 85 | 3.26 | 0.53 | 0.35 | 2.57 | 0.45 | 1.33 | 0.87 | 3.28 | 0.47 | 0.42 | 2.61 | 0.44 | 1.32 | 0.87 | 3.34 | 0.48 | 0.41 |
| 9 | 54306 | 2.90 | 0.32 | 1.34 | 0.87 | 2.96 | 0.58 | 0.38 | 2.49 | 0.51 | 1.28 | 0.86 | 3.01 | 0.53 | 0.44 | 2.54 | 0.50 | 1.18 | 0.90 | 3.18 | 0.51 | 0.43 |
| 8 | 55011 | 3.04 | 0.27 | 1.37 | 0.86 | 3.07 | 0.57 | 0.34 | 2.73 | 0.43 | 1.30 | 0.86 | 2.97 | 0.53 | 0.45 | 2.73 | 0.44 | 1.39 | 0.85 | 2.92 | 0.59 | 0.39 |
| 7 | 56113 | 2.86 | 0.33 | 1.11 | 0.90 | 2.91 | 0.61 | 0.35 | 2.55 | 0.44 | 1.18 | 0.88 | 3.01 | 0.58 | 0.48 | 2.57 | 0.44 | 1.09 | 0.91 | 2.88 | 0.64 | 0.45 |
| 6 | 57718 | 2.85 | 0.33 | 1.10 | 0.89 | **2.78** | **0.62** | **0.36** | 2.39 | 0.52 | 1.11 | 0.90 | **2.65** | **0.65** | **0.51** | 2.49 | 0.48 | 1.10 | 0.90 | **2.76** | **0.65** | **0.48** |
| 5 | 60338 | 2.75 | 0.36 | 1.15 | 0.89 | 3.00 | 0.56 | 0.34 | 2.51 | 0.45 | 1.17 | 0.88 | 2.88 | 0.55 | 0.48 | 2.53 | 0.45 | 1.07 | 0.90 | 3.00 | 0.55 | 0.39 |
| 4 | 65018 | 2.76 | 0.36 | 1.12 | 0.90 | 3.26 | 0.43 | 0.33 | 2.39 | 0.49 | 1.00 | 0.92 | 2.95 | 0.49 | 0.46 | 2.46 | 0.48 | 0.94 | 0.92 | 3.17 | 0.43 | 0.41 |
| 3 | 74575 | 2.64 | 0.41 | 0.97 | 0.91 | 3.47 | 0.46 | 0.34 | 2.19 | 0.59 | 0.98 | 0.92 | 2.88 | 0.59 | 0.48 | 2.22 | 0.58 | 0.86 | 0.94 | 2.74 | 0.62 | 0.46 |
| 2 | 98630 | 2.85 | 0.32 | 0.95 | 0.91 | 3.03 | 0.56 | 0.34 | 2.19 | 0.61 | 0.97 | 0.91 | 2.35 | 0.73 | 0.60 | 2.28 | 0.57 | 0.89 | 0.94 | 2.43 | 0.71 | 0.50 |
| 1 | 203379 | 2.62 | 0.41 | 0.85 | 0.93 | 3.31 | 0.52 | 0.37 | 2.15 | 0.60 | 0.90 | 0.94 | 2.16 | 0.81 | 0.69 | 2.20 | 0.58 | 0.85 | 0.95 | 2.35 | 0.78 | 0.61 |
| 0.9 | 235484 | 2.73 | 0.37 | 0.93 | 0.93 | 3.32 | 0.50 | 0.35 | 2.14 | 0.61 | 0.81 | 0.95 | 2.39 | 0.75 | 0.66 | 2.22 | 0.58 | 0.86 | 0.94 | 2.35 | 0.76 | 0.59 |
| 0.8 | 280084 | 2.75 | 0.36 | 0.90 | 0.94 | 3.10 | 0.54 | 0.34 | 2.22 | 0.58 | 0.90 | 0.94 | 2.00 | 0.84 | 0.73 | 2.25 | 0.56 | 0.78 | 0.95 | 2.02 | 0.84 | 0.66 |
| 0.7 | 345398 | 2.80 | 0.33 | 0.93 | 0.93 | 3.18 | 0.54 | 0.34 | 2.24 | 0.57 | 0.90 | 0.94 | 2.23 | 0.79 | 0.72 | 2.26 | 0.56 | 0.82 | 0.94 | 2.30 | 0.79 | 0.65 |
| 0.6 | 450939 | 2.81 | 0.34 | 0.93 | 0.92 | **3.21** | **0.54** | **0.33** | 2.18 | 0.60 | 0.86 | 0.94 | **2.11** | **0.81** | **0.70** | 2.20 | 0.59 | 0.85 | 0.94 | **1.99** | **0.84** | **0.63** |
| 0.5 | 631794 | 2.79 | 0.35 | 0.89 | 0.93 | 3.29 | 0.55 | 0.33 | 2.28 | 0.57 | 0.96 | 0.93 | 2.14 | 0.81 | 0.71 | 2.31 | 0.55 | 0.85 | 0.94 | 2.06 | 0.83 | 0.65 |
| 0.4 | 965994 | 2.86 | 0.31 | 0.93 | 0.93 | 3.44 | 0.53 | 0.35 | 2.43 | 0.52 | 0.96 | 0.93 | 2.31 | 0.82 | 0.66 | 2.39 | 0.52 | 0.89 | 0.93 | 2.27 | 0.82 | 0.65 |
| 0.3 | 1515513 | 2.83 | 0.32 | 0.96 | 0.93 | 3.30 | 0.58 | 0.36 | 2.12 | 0.65 | 0.84 | 0.95 | 3.10 | 0.65 | 0.53 | 2.16 | 0.63 | 0.81 | 0.95 | 3.18 | 0.67 | 0.50 |

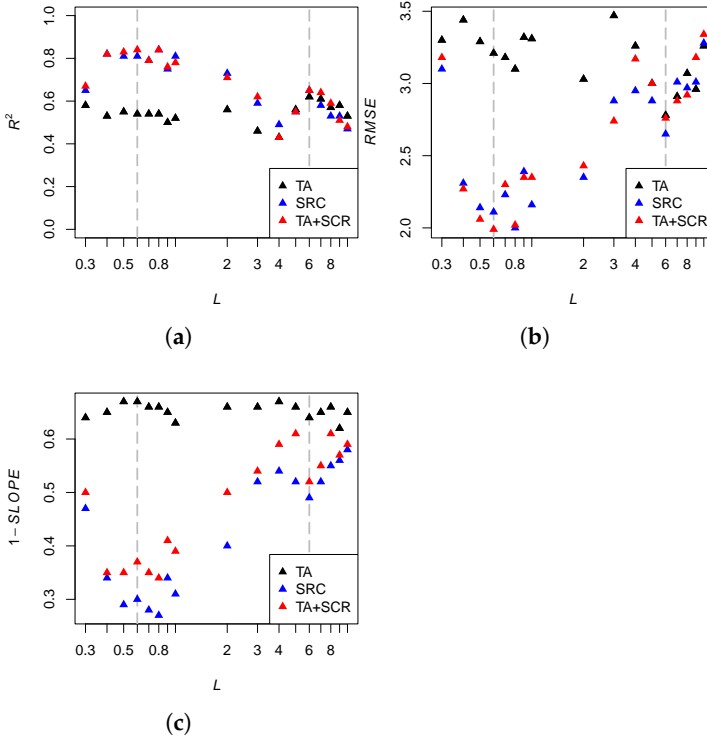

**Figure 7.** Relations between logarithmized scale level $L$ and accuarcy metrics $R^2_{VAL}$ (**a**), $RMSE_{VAL}$ (**b**), and $SLOPE_{VAL}$ (**c**) (cf., Table 3) for the parametrization variants *terrain attributes* (TA), *SCMaP-SRC* (SRC), and *terrain attributes + SCMaP-SRC* (TA + SRC). The gray dashed line refers to scale levels $L = 0.6$ and $L = 6$ (cf., Figures 8 and 9).

Each scale level is characterized by scatterplots of the predicted and observed SOC values with respect to the test data. Figure 8 illustrates this for all three parametrization variants and the scale levels $L = 0.6$ and $L = 6$, which can be considered as representatives of both scale ranges. Besides the measures $R^2$ and $RMSE$, the slope of the regression line ($SLOPE$) as an indicator for model over- or underestimation is displayed. According to Figure 7, all models lead to a SOC underestimation. The degree of underestimation is the lowest for the SRC-based models in the scale range between $L = 0.4$ and $L = 1$. It is also noticeable that the pure SRC models show better results than the mixed TA and SRC models. Both effects can be observed in Figure 8.

The most accurate modeling variant regarding $R^2$ and $RMSE$ with "scale level $L = 0.6$, parametrization variant C" (Figure 8f) is mapped in Figure 9. The SOC content pattern reflects the soil and terrain landscape structure with higher SOC values in lowlands and lower SOC values in hilly regions (cf., Figure 4). This is also true for the prediction "scale level $L = 6$, parametrization variant A" (Figure 8a), which makes the main landscape structures visible. However, a visual comparison of both variants shows more detailed differentiation and higher spatial variability of predicted SOC contents for the $L = 0.6$ variant, which is particularly pronounced in the lowlands. This is also where the greatest differences in the value distributions between the two variants can be observed, which lie roughly in the value range between 4 and 8% SOC content (Figure 10). Furthermore, a comparison of the two SOC content distributions with the distribution of the training data set shows that the $L = 6$ prediction variant deviates more than the $L = 0.6$ variant, as shown by the Kolmogorov–Smirnov (KS) distances of the corresponding empirical cumulative distribution functions (ECDF) (cf., [6]).

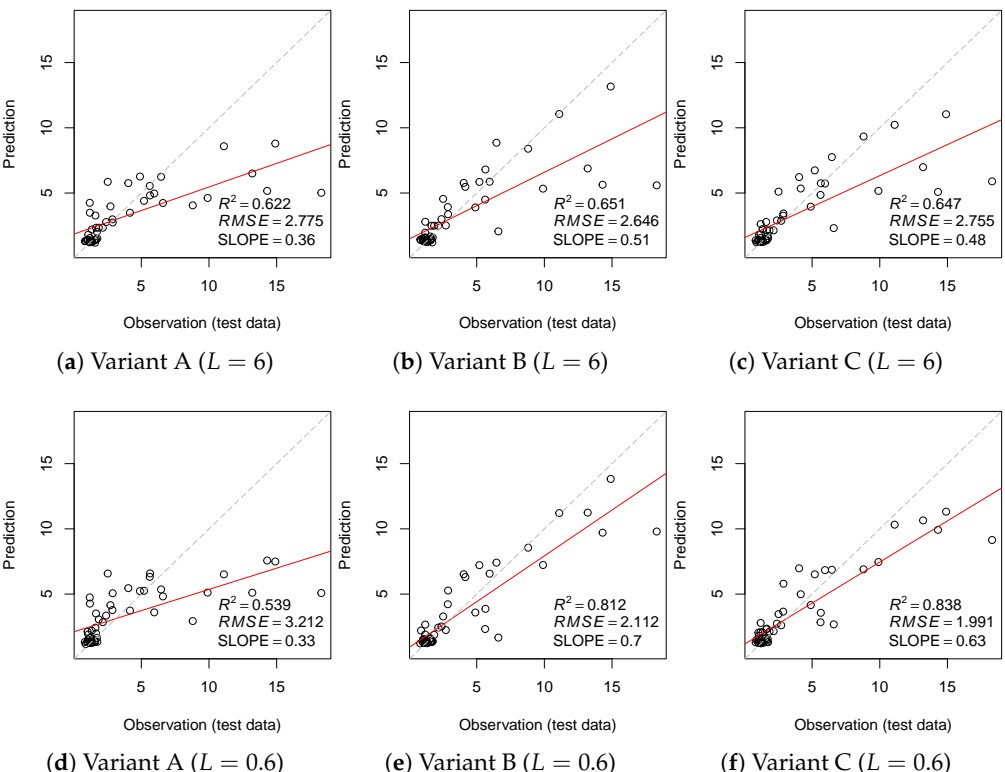

(**a**) Variant A (*L* = 6)  (**b**) Variant B (*L* = 6)  (**c**) Variant C (*L* = 6)

(**d**) Variant A (*L* = 0.6)  (**e**) Variant B (*L* = 0.6)  (**f**) Variant C (*L* = 0.6)

**Figure 8.** Comparison of SOC content (%) validation results based on test data sets for six prediction variants (cf., Table 3). The red line is the regression line of the scatter plot, the gray and dashed line indicates the optimal regression line.

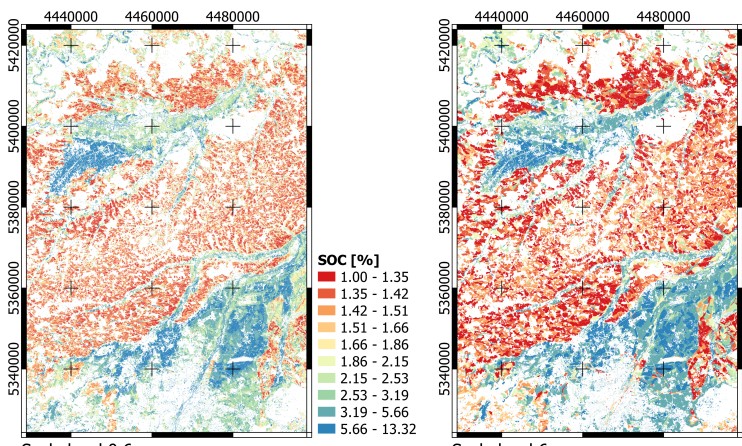

**Figure 9.** Predicted SOC content (%) values for scale level *L* = 0.6 and parametrization variant *terrain attributes + SCMaP-SRC* (TA+SRC) (left) as well as scale level *L* = 6 and parametrization variant *terrain attributes* (TA) (cf., Table 3 and Figure 8a,f). Projection: EPSG 31468.

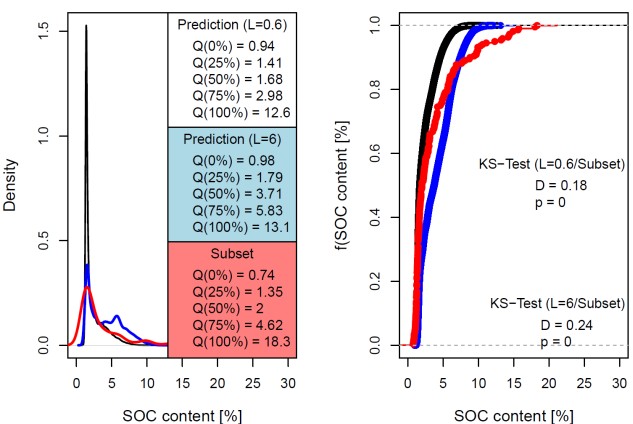

**Figure 10.** Comparison of soil samples' (red) and predicted SOC content (%) distributions (black/blue): density plots with quantile values $Q$ (**left**) and plots of empirical cumulative distribution functions (ECDF) with Kolmogorov–Smirnov (KS) distances $D$ (**right**) for two prediction variants.

Table 4 summarizes the recursive feature elimination results. There, the most relevant parameters, which lead to minimal $RMSE$ values, are listed. In addition to the one-dimensional terrain features $FILL$, $SLP$, $TOP$, and $TON$, variants of the multi-hierarchical terrain features $NH$ and (subordinately) $VDC$ have the greatest influence on the modeling results. As for the SRC attributes, in particular the normalized multi-temporal Landsat-band 5 ($SRC_{12}$) as well as the bands 2 ($SRC_{2,9}$) and 3 ($SRC_{3,10}$) are important. The analysis of the combined parameterization variants reveals the dominance of the SRC attributes at all scale levels. Figure 11 visualizes the example of two modeling variants that, with only a few attributes, lead to a significant reduction of $RMSE$ values.

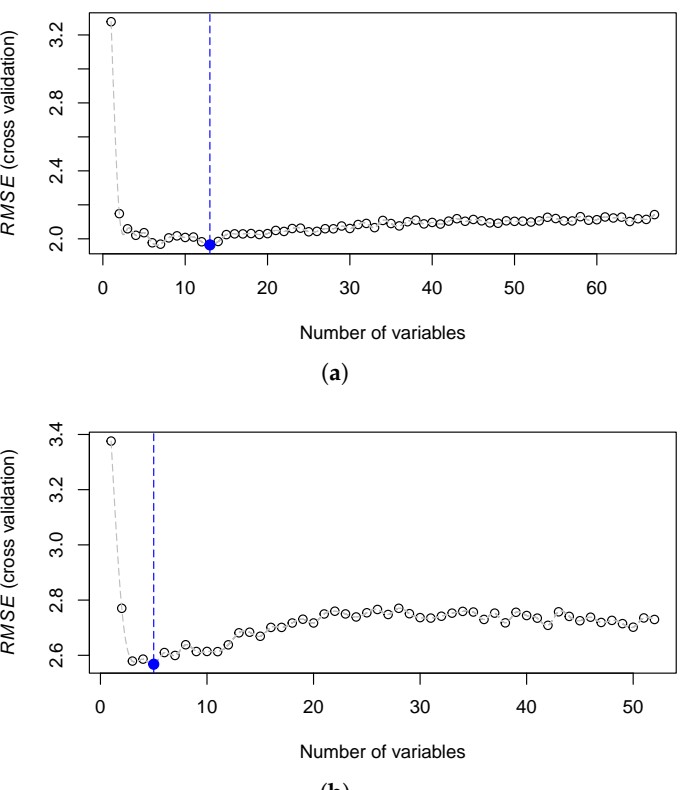

**Figure 11.** RFE-based dependencies between $RMSE$ values and parameter combinations for scale level $L = 0.6$ and parametrization variant *terrain attributes + SCMaP-SRC* (TA+SRC) (**a**) as well as scale level $L = 6$ and parametrization variant *terrain attributes* (TA) ((**b**); Table 4). The dashed blue lines indicate minimal $RMSE$ values.

**Table 4.** The most important parameters based on RFE algorithm, which lead to minimal *RMSE* accuracy metrics. The order of the parameters represents their meaning. The gray marked table cells refer to Figure 11.

| Scale Level | TA | SRC | TA+SRC |
|---|---|---|---|
| 10.0 | $FILL, TOP, SLP, MBI_1$ | $SRC_{10}, SRC_{12}, SRC_3, SRC_1, SRC_9, SRC_4, SRC_2, SRC_{11}, SRC_{15}, SRC_7$ | $SRC_{10}, SRC_9, SRC_{12}, SRC_3, SRC_2, SRC_{11}, SRC_4, SRC_1, NH_2, SRC_{15}, SRC_7$ |
| 9.0 | $FILL, TOP, VDC_{599484}, MBI_1$ | $SRC_{10}, SRC_{12}, SRC_2, SRC_7, SRC_3, SRC_9, SRC_{11}, SRC_5, SRC_4, SRC_1, SRC_{13}, SRC_6$ | $SRC_{10}, SRC_2, SRC_9, SRC_3, SRC_{11}, SRC_{12}, SRC_7, SRC_{15}, SRC_4$ |
| 8.0 | $FILL, NH_4, TOP$ | $SRC_{10}, SRC_{12}, SRC_7, SRC_3, SRC_2$ | $SRC_{10}, SRC_3, SRC_2, SRC_9, SRC_{12}$ |
| 7.0 | $TOP, FILL, TON$ | $SRC_{10}, SRC_3, SRC_{12}, SRC_4$ | $SRC_3, SRC_{10}, SRC_4, SRC_{12}, SRC_2, SRC_{15}, VDC_{46416}, SRC_1, SRC_9, TON, TOP, SRC_5, SRC_7, VDC_{359381}, SRC_{11}, FILL$ |
| 6.0 | $FILL, TOP, NH_4, SLP, VDC_{46416}$ | $SRC_{12}, SRC_3, SRC_{10}$ | $SRC_3, SRC_{12}, SRC_2, SRC_{10}, SRC_{15}$ |
| 5.0 | $FILL, NH_4, TOP, SLP, NH_8$ | $SRC_{12}, SRC_3, SRC_{10}$ | $SRC_3, SRC_{15}, SRC_4, SRC_2, SRC_{10}, SRC_{12}, FILL, SRC_9, SRC_{13}, SRC_7, SRC_1, TOP, SRC_{11}, SLP, NH_4$ |
| 4.0 | $FILL, TOP, NH_4, NH_8$ | $SRC_3, SRC_{12}, SRC_{10}, SRC_{15}, SRC_1$ | $SRC_3, SRC2, SRC_{15}, SRC_{12}, SRC_{10}, SRC_4, SRC_1, SRC_9, SRC_7, SRC_5, SRC_{13}, SRC_{11}$ |
| 3.0 | $FILL, SLP$ | $SRC_{12}, SRC_3, SRC_{10}, SRC_4, SRC_2, SRC_{11}, SRC_{15}, SRC_9, SRC_1$ | $SRC_3, SRC_{12}, SRC_{10}, SRC_2, SRC_4, SRC_{15}, SRC_9, SRC_{11}, SRC_1, SRC_5, SRC_{13}, SLP, VDC_{359381}, TON, SRC_7, VDC_{16681}$ |
| 2.0 | $FILL, NH_4$ | $SRC_{12}, SRC_3, SRC_{10}, SRC_2, SRC_{14}, SRC_4, SRC_9, SRC_1, SRC_5, SRC_{13}$ | $SRC_3, SRC_2, SRC_{12}, SRC4, SRC_5, SRC_{10}, SRC_1, SRC_9, SRC_{15}, SRC_{14}, SRC_7, SRC_{13}, FILL, VDC_{10000}, SLP, SRC_{11}, VDC_{599484}, VDC_{359381}$ |
| 1.0 | $FILL, NH_4, SLP$ | $SRC_{12}, SRC_{10}, SRC_2, SRC_3, SRC_4, SRC_9, SRC_{11}, SRC_{14}, SRC_8, SRC_1, SRC_5, SRC_{13}, SRC_7, SRC_6$ | $SRC_{12}, SRC_2, SRC_3, SRC_4, SRC_{10}, SRC_{14}, SRC_5, SRC_9, SRC_1, SRC_{13}, SRC_{11}, FILL, SRC_{15}, NH_4, VDC_{129155}, SLP$ |
| 0.9 | $FILL, NH_4$ | $SRC_{12}, SRC_2, SRC_{10}, SRC_3, SRC_9, SRC_4, SRC_{11}, SRC_1, SRC_{14}$ | $SRC_{12}, SRC_2, SRC_3, SRC_{10}, SRC_4, SRC_9, VDC_{129155}, SRC_1, SRC_{14}, SRC_5$ |
| 0.8 | $FILL, NH_4$ | $SRC_{12}, SRC_{10}, SRC_2$ | $SRC_{12}, SRC_2, SRC_3, SRC_{10}$ |
| 0.7 | $FILL, NH_4$ | $SRC_{12}, SRC_{10}, SRC_2$ | $SRC_{12}, SRC_2, SRC_3, SRC_{10}, SRC_4, SRC_{13}$ |
| 0.6 | $FILL, NH_4$ | $SRC_{12}, SRC_2, SRC_{10}, SRC_3, SRC_9, SRC_4, SRC_{13}, SRC_8, SRC_{11}, SRC_5$ | $SRC_{12}, SRC_2, SRC_3, SRC_{10}, SRC_4, SRC_{13}, SRC_9, VDC_{129155}, SRC_5, NH_4, SRC_1, SRC_{15}, FILL$ |
| 0.5 | $FILL, NH_4, TON$ | $SRC_{12}, SRC_2, SRC_3, SRC_{10}$ | $SRC_2, SRC_{12}, SRC_3, SRC_{13}, SRC_{10}, SRC_4, FILL, VDC_{129155}$ |
| 0.4 | $FILL, NH_4$ | $SRC_{12}, SRC_2, SRC_3, SRC_{10}$ | $SRC_2, SRC3, SRC_{12}$ |
| 0.3 | $FILL, NH_4, TON$ | $SRC_2, SRC_{12}, SRC_{10}, SRC_3, SRC_9, SRC_{13}, SRC_7, SRC_1, SRC_8, SRC_4$ | $SRC_2, SRC_3, SRC_{13}, SRC_{12}, VDC_{359381}, SRC_4, SRC_{10}, TON, NH_2, SRC_1, FILL, NH_4, SRC_7, SRC_5, VDC_{10000}, SLP, SRC_{11}, VDC_{46416}$ |

## 4. Discussion

### 4.1. Data Quality and Fitness-for-Use

Approved data quality is a prerequisite for the reusability of data. The SOC content modeling results presented are examples of standardized soil mapping products. Standardization refers to reproducible data processing and modeling, as well as their evaluation based on accuracy metrics [6,69–71]. In contrast to static conventional soil maps, the scale-specific suitability can be determined, which helps to communicate map quality to end-users [71–73], to provide additional information about data fitness-for-use [74,75], to improve the model's interpretability [4] as well as to get a additional geospatial provenance description [76]. Although the process chain presented is reproducible, individual steps are based on expert knowledge. This concerns in particular the selection of segmentation parameters (Section 2.1.4) and terrain attributes as well as their multi-scale tuning parameters (Table 4). Here, further research is needed to define statistically sound parameters [25,67].

The validation scheme in this study follows the approach of Zepp et al. (2021) [5] and the recommendations of Piikki et al. (2021) [73] including data splitting, cross-validation, and independent validation, as well as the use of different types of accuracy metrics. The latter were primarily used to compare different scale-specific parameterization variants. Maps for practical use should also contain uncertainty metrics, which estimate the prediction variation for every raster cell. Geostatistical metrics or prediction intervals (e.g., [18,61,77–79]) are widely used.

### 4.2. Scale-Specific Optimization

The SOC content map quality is affected by factors such as the spatial and temporal representativeness of the samples or the scale-specific explanatory power of the variables used. Following the effective map scale (EMS) approach [30], each scale-specific map is characterized by its "predictive efficiency" [33]. The underlying workflow can be considered as a procedure where the relationship between SOC content samples and different aggregation levels of multi-scale terrain attributes and SCMaP-SRCs is statistically optimized. This is also evident in the comparison of the modeling results based on the filter-specific parameterization (Section 3.1), which represents a static window-based aggregation procedure. In contrast to changes in grid resolution of terrain attributes [26,28], the segmentation-based aggregation considers both parameter-specific and spatial data variability. In this way, a more precise delineation of the reference units can be made. This is relevant, for example, for samples taken at field boundaries [5]. This means that the optimization can counteract possible positional inaccuracies of the samples [27,80].

The accuracy measure $R^2$ of the best modeling SCMaP-SRC variant ($L = 0.6$, variant B with $R^2 = 0.81$; Table 3) exceeds the result of Zepp et al. (2021) ($R^2 = 0.74$; Table 2) [5], whereas the $RMSE$ values are the same for both models ($RMSE = 2.11$). It can be assumed that the accuracy measures of the SCMaP-SRC variant $L = 0.3$ (variant B), which corresponds to the original raster resolution, are exceeded by both variants of the filter- and scale-specific parameterization models. The additional consideration of terrain attributes leads to a further model improvement regarding both accuracy measures ($L = 0.6$, variant C with $R^2 = 0.84$ and $RMSE = 1.99$; Table 3).

The prediction results made a jump in scale visible (cf., Figure 7). They refer to concepts of hierarchical landscape structuring, according to which (here: soil-relevant) processes and states are associated with specific scale ranges [30,81,82]. Accordingly, SCMaP-SRC-related accuracy measures in particular show significant differences around the $L = 3$ level, with almost the same spectral bands having the highest impact on predictions at all scale levels. Compared to terrain attributes, SCMaP-SRC parameters are also characterized by a higher explanatory power at fine scales, especially below scale level $L = 3$.

TA-related accuracy measures display a smaller and more balanced variation across scale levels. One reason might be that various expressions of terrain attributes have been used as explanatory variables. They represent variations regarding scale or terrain complexity. This means that in addition to scale optimization, the terrain attribute variations

also serve as optimization variables [30]. Of the multi-scale terrain attributes, the $NH$ and the $VDC$ variants are particularly relevant at different scale levels, which Guo et al. (2019) also consider as key attributes that influence SOC distribution [26]. While below scale level $L = 3$ the multi-scale attribute variant $NH_4$ dominates, above scale level $L = 3$ other $NH$ or $VDC$ variants appear. This underlines the scale dependence of the soil-related processes, for which scale-specific parameters have to be identified as optimal for prediction [28,83,84]. It is finally noticeable that the one-dimensional attribute $FILL$ has the highest explanatory power. Other one-dimensional attributes of high relevance are $SLP$ and $TON/TOP$.

From a machine learning perspective, all used explanatory variables represent "hand-crafted" features whose selection is based on domain or expert knowledge [85]. This mainly concerns the determination of multi-scale tuning parameters (cf., Table 1) and scale levels (see Section 2.1.4). Although reproducible, there is potential for unsupervised and statistically driven approaches for the derivation of the parameters (e.g., [56,86]). This is also true for object-based contextual parameters [56], which have not been considered in this study.

*4.3. SCMaP-SRC as Additional Input for SOC Modeling*

The results shown in Section 3.2 indicate an increase in the SOC model performances using SCMaP-SRC data in addition to terrain attributes (see Figure 7). Though the $R^2$ values (both model calibration and validation) for TA and TA+SRC point to high model performances, the $RMSE$ are relatively high (>1.99). The federal state of Bavaria shows a wide range of SOC contents, as mineral and organic soils occur. As the focus of the subset definition was on the selection of a representative subarea of the entire federal state, a possible wide range of SOC contents was included here. Hence, the high $RMSE$ could be related to the wide range of SOC contents in the study area. Relatively high $RMSE$ values for SOC modeling in Bavaria were also reported by Zepp et al. (2021) [5]. According to the results of the recursive feature elimination shown in Table 4, the most important SRC attributes are bands 2 and 3 which are selected over all different scale levels. Zepp et al. (2021) also showed the importance of the bands 2 and 3 for the SOC modeling [5]. Additionally, band 12 (band 5 normalized) is of high importance for SOC modeling.

To investigate the influence of the combination of terrain attributes and the SCMaP-SRC information, the same remote sensing database as shown in Zepp et al. (2021) [5] was used. The SOC modeling was performed using a spatial subset of the 30-year SCMaP SRC data. The 30-year compositing period enables stable conditions and mainly includes permanent spatial soil moisture differences, related to soil texture or type characteristics. Influencing factors as varying short-term soil moisture differences thus have a lower effect compared to analysis based on shorter compositing lengths or single scenes. However, an analysis of this assumption is still necessary. Additionally, a long compositing period enables the integration of a high number of cloudless scenes, which is accountable for a reliable data source [49]. Here, the 30-year period was applicable, as, among others shown by Kühnel et al. (2020) [87], SOC contents are constant for Bavaria. Based on permanent observation sites, no to low SOC changes were observed between 1984 and 2016. However, the use of a 30-year composite could hamper the SOC prediction if the investigation area includes short-term SOC changes or changes over several years. For the transferability of the shown modeling techniques to other areas with temporally higher SOC changes, shorter compositing periods have to be considered. In addition, an investigation of the impacts of political regulations (e.g., carbon farming [88] or denser modeling of soil health and status) would be enabled. The integration of Sentinel-2 data can potentially shorten the compositing time length, as the twin satellites provide a huge amount of data based on the combined revisit time of fewer than five days [89,90]. Additionally, the global available Harmonized Landsat Sentinel-2 (HSL) surface reflectance data set [91] can be considered. Both harmonized data sets are based on the same pre-processing schemes, enabling the data set as a highly valuable input regarding the large number of available scenes for the compositing approach.

## 5. Conclusions

In this study, approaches of multi-scale feature engineering, geographic object image analysis (GEOBIA), and machine learning have been coupled to a workflow where relations between SOC content samples as well as different aggregation levels of multi-scale terrain attributes and multi-temporal soil reflectance composites are optimized. The main findings of the study can be summarized as follows:

- There are scale-specific dependencies between the representativeness of the soil samples and the explanatory power of the variables used.
- Compared to terrain attributes, parameters based on multi-temporal soil reflectance composites are characterized by a higher explanatory power at fine scales.
- The explanatory power of terrain attributes is generally smaller but more balanced across scale levels.
- The best modeling variant is characterized by an accuracy of $R^2 = 0.84$ and $RMSE = 1.99$, which outperforms modeling results based on a static window-based aggregation procedure with $R^2 = 0.74$ and $RMSE = 2.11$.
- The study results suggest that DSM workflows should include scale-related optimizations.

**Author Contributions:** Conceptualization, M.M. and U.H.; methodology, M.M. and S.Z.; software, M.M.; validation, M.M. and M.W.; formal analysis, M.M.; investigation, M.M.; data curation, M.M., S.Z. and M.W.; writing—original draft preparation, M.M. and S.Z.; visualization, M.M.; supervision, M.M., U.H. and H.G.; project administration, H.G.; funding acquisition, U.H. and H.G. All authors have read and agreed to the published version of the manuscript.

**Funding:** This research was funded by the German Federal Ministry of Food and Agriculture (BMEL), grant number 281B301816 as part of the Soil-DE project "Entwicklung von Indikatoren zur Bewertung der Ertragsfähigkeit, Nutzungsintensität und Vulnerabilität landwirtschaftlich genutzter Böden in Deutschland".

**Data Availability Statement:** Not applicable.

**Acknowledgments:** We thank the Bavarian agencies, the Bavarian Environment Agency (LfU) and the Bavarian State Research Center for Agriculture (LfL) for providing the soil databases.

**Conflicts of Interest:** The authors declare no conflict of interest. The funders had no role in the design of the study; in the collection, analyses, or interpretation of data; in the writing of the manuscript, or in the decision to publish the results.

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
