# Peer review of "Scale-Specific Prediction of Topsoil Organic Carbon Contents Using Terrain Attributes and SCMaP Soil Reflectance Composites"

_remotesensing, doi:10.3390/rs14102295_

Round 1

Reviewer 1 Report

Authors in the manuscript deal with approaches of multi-scale feature engineering, geographic object image analysis and machine learning which have been coupled to a workflow where relations between soil organic carbon content samples as well as different aggregation levels of multi scale terrain attributes optimized. From the conclusions is clear that there are scale-specific dependencies between the representativeness of the soil samples and the explanatory power of the variables used. Compared to terrain attributes, parameters based on multi-temporal soil reflectance composites are characterized by a higher explanatory power at fine scales. The explanatory power of terrain attributes is generally smaller but more balanced across scale levels. According my opinion the manuscript is written on a very good level. I consider the conclusions in the manuscript are original. I did not find any serious defects in the work or in the presentation or ethical problems. I appreciate the large amount of studied literature /89 references listed/ which testifies to the good orientation of the authors in the given issue. In my opinion, the keywords are consistent with the content of the article. I do not have any serious comments and manuscript „Scale-specific Prediction of Topsoil Organic Carbon Contents using Terrain Attributes and SCMaP Soil Reflectance Composites” can be published in the Remote sensing.

Author Response

Thank you for your positive and motivating review of the manuscript.

Reviewer 2 Report

see attached document

Author Response

Thank you for your positive and motivating review of the manuscript. In the following, we address your comments and suggestions: 

Point 1: A few words can be added to explain the choice of this test site out of all the different soil samples in Bavaria.

The size of the test area initially had pragmatic reasons related to the capacity of the segmentation algorithm. However, as stated in chapter 2.1.1, care was taken in the selection of the area to ensure that the value ranges and soil relief ratios were as comparable as possible to those of Bavaria (Fig. 3). In addition, comparative modeling was performed to ensure the comparability of this study's results in particular with those of Zepp et al. (2021) (see Table 2).

Point 2: My interrogation is about the time period used (1984-2014). Why stop in 2014? Are the following data available? If so, why not include them in the document, choosing a more contemporary period, from 1990-2020 for example? 

We have added an information in section 2.3.3 (Line 134): "The compositing period was chosen according to the dates of soil sampling. " In section 4.3, we discuss the advantage of the modeling period, which is mainly to minimize the influence of varying soil moisture.

Point 3: p. 2 line 73 “Maschine larning”. Is it correct?

Corrected.

Point 4: p. 6 lines 128 - 134 Due to the many data processing/filtering steps, can the authors give indications on the ratio of final data used to initial raw data?

The numbers of available and used scenes have been added (Lines 135-136): "The 30 year SRC was built on all Landsat (69 Landsat-4 TM, 1784 Landsat-5 ETM and 998 Landsat-7 ETM+) collection scenes \cite{wulder_current_2019} available between 1984 and 2014 with an resolution of 30 m for the investigation area." 

Point 5: p. 7 figure 5 The dashed yellow line is not clearly visible. Another color will be more suitable.

The yellow line color was replaced by orange, which should be now more visible.

Point 6: p. 10 lines 255 Figure 8 is introduced in the text after Figure 9!

We have corrected the order of figures.

Reviewer 3 Report

Manuscript ID: remotesensing-1658916
Type of manuscript: Article
Title: Scale-specific Prediction of Topsoil Organic Carbon Contents using
Terrain Attributes and SCMaP Soil Reflectance Composites
Authors: Markus Möller *, Simone Zepp, Martin Wiesmeier, Heike Gerighausen, Uta Heiden
Submitted to section: Environmental Remote Sensing,
Remote Sensing for Soil Organic Carbon Mapping and Monitoring

A brief summary

The article is of great interest. Abstract is well articulated. The goals set have been achieved. The methods are described fully enough to reproduce them. The literature reflects the world level of research on the problem. The reviewer notes the high quality of the preparation of the Manuscript.

The manuscript can be recommended for publication.

However, the Reviewer would like to point out one shortcoming.

The reviewer believes that according to the data from section 2.1.2. there is a methodological problem. It concerns the lack of an assessment of sample homogeneity in terms of SOC values. The authors write that “The SOC contents range from 0.74% to

18.3% with a median content of 2.00%.’ ​​(L 91-92). The reviewer asks the authors to name those soil types (subtypes), according to WRB, that characterize a sample of 220 soil samples. And, if among these soils, in addition to automorphic (mineral) soils, there are hydromorphic soils, which had values ​​up to 18.3% SOC, i.e. 32% OM (humus), then they, of course, cannot be considered together with automorphic (mineral) soils. This is, as they say, "a tail from another cat."

 Specific Comments:

References

Authors should unify the Title of cited articles. That is, change Capital letters to lowercase for sources No. 4, 5,  12-14, 31, 33, 40, 42,56, 64, 65, 73, 86, 88.

Author Response

Thank you for your positive and motivating review of the manuscript. In the following, we address your comments and suggestions: 

Point 1: The reviewer believes that according to the data from section 2.1.2. there is a methodological problem. It concerns the lack of an assessment of sample homogeneity in terms of SOC values. The authors write that “The SOC contents range from 0.74% to 18.3% with a median content of 2.00%.’ ​​(L 91-92). The reviewer asks the authors to name those soil types (subtypes), according to WRB, that characterize a sample of 220 soil samples.

We have added a sentence, which characterizes the mail soil formations of the test site according to WRB nomenclature (Lines  81-86): "The data set comprises soils with well-developed B horizons (mainly Cambisols), soils with initial soil formation (mainly Leptosols), soils with water stagnation (mainly Stagnosols and Planosols), soils with clay migration (mainly Luvisols), clay-rich soils (mainly Vertisols), groundwater soils (mainly Gleysols) and natural bogs and fens (mainly Histosols) according to the German soil systematic and the equivalent Reference Soil Groups of the WRB system (IUSS Working Group WRB, 2015)."

Point 2: And, if among these soils, in addition to automorphic (mineral) soils, there are hydromorphic soils, which had values ​​up to 18.3% SOC, i.e. 32% OM (humus), then they, of course, cannot be considered together with automorphic (mineral) soils. This is, as they say, "a tail from another cat."

You correctly noted that mineral and hydromorphic soil samples were analyzed together in this study. We followed this approach because we believe that especially the decision tree method like the Random Forest algorithm can handle the data situation. The prerequisite is that explanatory variables exist that can be used to characterize the two different soil formation conditions. Especially for the distinction between mineral and hydromorphic (especially groundwater-influenced) soil formation conditions, relief attributes, but increasingly also reflectances have proven to be useful. The challenge here is to define multi-scale relief attributes, since hydromorphic soils in particular occur at different scales such as floodplains and depressions (cf. Möller et al. 2008).

Möller, M., Volk, M., Friedrich, K., Lymburner, L., 2008. Placing soil-genesis and transport processes into a landscape context: A multiscale terrain-analysis approach. J. Plant Nutr. Soil Sci. 171, 419–430. https://doi.org/10.1002/jpln.200625039

Point 3: Authors should unify the Title of cited articles. That is, change Capital letters to lowercase for sources No. 4, 5,  12-14, 31, 33, 40, 42,56, 64, 65, 73, 86, 88.

Corrected.